# An Improved Method for Broiler Weight Estimation Integrating Multi-Feature with Gradient Boosting Decision Tree

**DOI:** 10.3390/ani13233721

**Published:** 2023-12-01

**Authors:** Ximing Li, Jingyi Wu, Zeyong Zhao, Yitao Zhuang, Shikai Sun, Huanlong Xie, Yuefang Gao, Deqin Xiao

**Affiliations:** 1College of Mathematics and Informatics, South China Agricultural University, Guangzhou 510642, China; 2Key Laboratory of Smart Agricultural Technology in Tropical South China, Ministry of Agriculture and Rural Affairs, Guangzhou 510642, China; 3School of Computer & Communication Engineering, University of Science and Technology Beijing, Beijing 100083, China; 4Wens Foodstuff Group Co., Ltd., Yunfu 527400, China

**Keywords:** broiler, depth image, instance segmentation, weight estimation, gradient boosting decision tree

## Abstract

**Simple Summary:**

In the broiler farming industry, accurately weighing broilers is essential. Camera-based weighing systems offer a solution without requiring expensive platform weighers. However, existing computer vision methods for estimating broiler weight are limited to young broilers and become less accurate as broilers age due to increased density and weight imbalance. To tackle this, a new approach is introduced in this paper. It uses a segmentation network (Mask R-CNN) with depth images captured by a 3D camera to isolate broilers. Artificial and learned features are combined using a feature fusion module, and these features are used to estimate broiler weight through gradient boosting decision trees (GBDT). The method excels in accurately estimating the weight of individual broilers within complex scenes and holds promise for enhancing broiler weight estimation methods.

**Abstract:**

Broiler weighing is essential in the broiler farming industry. Camera-based systems can economically weigh various broiler types without expensive platforms. However, existing computer vision methods for weight estimation are less mature, as they focus on young broilers. In effect, the estimation error increases with the age of the broiler. To tackle this, this paper presents a novel framework. First, it employs Mask R-CNN for instance segmentation of depth images captured by 3D cameras. Next, once the images of either a single broiler or multiple broilers are segmented, the extended artificial features and the learned features extracted by Customized Resnet50 (C-Resnet50) are fused by a feature fusion module. Finally, the fused features are adopted to estimate the body weight of each broiler employing gradient boosting decision tree (GBDT). By integrating diverse features with GBTD, the proposed framework can effectively obtain the broiler instance among many depth images of multiple broilers in the visual field despite the complex background. Experimental results show that this framework significantly boosts accuracy and robustness. With an MAE of 0.093 kg and an R^2^ of 0.707 in a test set of 240 63-day-old bantam chicken images, it outperforms other methods.

## 1. Introduction

It is economically important to obtain the weight information of broilers in chicken farms, as weight is a crucial indicator of broiler growth. In intelligent poultry farms, broiler weight is closely related to broiler yield, which most directly reflects their growth. Meanwhile, it is essential to monitor, record, and predict broilers’ body weight for rational intervention in their feed intake and health management, so that the optimal market timing can be determined. In addition, broiler weight information can be used to fit the growth curve, reveal the growth pattern of broilers, and estimate the weight uniformity for efficient flock management. Correspondingly, broiler weight information is an indispensable statistic in optimizing feed-to-meat ratio, selecting proper feed, and facilitating disease prevention and control during the breeding process. Therefore, it is vital to estimate the weight of every individual broiler in a quick and harmless way. Based on different approaches to collection and measurement, broiler weighing can be roughly divided into two categories: traditional methods and computer vision-based methods.

Traditional methods utilize direct manual weighing, during which a single broiler is repeatedly picked and placed on hanging scales, electronic scales, floor scales, or body weight boxes, and the average value obtained from the electronic scale is the body weight of a single broiler. Simple as they may appear, the traditional methods are both time-consuming and laborious. Moreover, some traditional weight prediction methods are based on the manual collection of morphometric measurements, which often involves many features including beak length, body length, keel length, thoracic circumference, toe length, body girth, shank length, back length, shank circumference, and wing length. These morphometric measurement data have to be manually selected and then used as features for traditional regression models or machine learning models such as Random Forest (RF) and Classification and Regression Tree (CRT) [1]. The manual selection process not only takes a long time but also increases the broiler’s stress reaction, making them languid in the subsequent breeding environment due to close contact with humans in weighing. In severe cases, the broiler will even be emotionally disturbed. Therefore, traditional methods are not conducive to animal welfare [2,3,4,5].

Computer vision-based methods applied in broiler farming can be traced back to 2003. De Wet et al. [6] explored the surface area and periphery of the broilers as weight descriptors. Mollah et al. [7] developed a linear equation to predict broilers’ weights from their body surface area pixels during the 42-day rearing period. Mortensen et al. [8] used 3D computer vision and 1D, 2D, and 3D features to predict broiler chicken weight. Likewise, Amraei et al. [9]; Amraei et al. [10]; and Amraei et al. [11] estimated broiler chicken weight using 2D features with SVR, ANN methods, and transfer function model, respectively. Wang et al. [12] extracted nine features for backpropagation neural network model construction. These studies suggest that camera-based weighing of broilers is indeed feasible. The computer vision-based methods, which utilize cameras to capture broiler pictures to estimate body weight, are a class of indirect methods characterized by a unified idea.

To realize data acquisition, a top view image of the broiler is collected by a 2D or 3D camera [13]. Using a 2D SAMSUNG digital camera (SM-N9005, Korea), Amraei et al. [9], Amraei et al. [10], and Amraei et al. [11] captured chicken broiler images in their research, while Mortensen et al. [8] and Wang et al. [12] employed Microsoft Kinect depth cameras to collect 3D image data for weight estimation of free-cage broilers.

Generally speaking, a computer vision-based method consists of three stages: the image segmentation stage, the feature extraction stage, and the weight estimation stage. First, in the image segmentation stage, Mortensen et al. [8] used the marker-based watershed segmentation technique to partition the broiler depth image with clean and clear background into several parts. Wang et al. [12] applied a threshold-based algorithm to separate the target broiler from depth image with a single broiler. Similarly, Amraei et al. [9], Amraei et al. [10], and Amraei et al. [11] directly adopted threshold-based algorithms on RGB images to obtain broiler instances. But this method will not work if the broiler chickens do not have white feathers. In brief, none of these segmentation methods can perfectly solve the problems of image depth with complex backgrounds and the inaccurate segmentation of multiple broilers images. Furthermore, it is difficult to keep a balance between segmentation effect and processing efficiency.

Second, in the feature extraction stage, Mortensen et al. [8] extracted thirteen features, including one-dimensional, two-dimensional, and three-dimensional ones from the target image and depth image, while Wang et al. [12] extracted nine features and Amraei et al. [10] extracted six features, respectively. These artificial features, extracted from the target broiler image with its head image removed, usually including area, perimeter, eccentricity, major axis length, and minor axis length, cause information loss. And there exists another problem when age is used as a one-dimensional feature to estimate a broiler’s weight. In practice, the high degree of similarity in body weight and the presence of outliers make it more challenging to estimate the weight of a vast number of broiler chickens of the same age. In addition, a low number of features will greatly increase the difficulty of broiler weight estimation and the complexity of the model.

Third, in the body weight estimation stage, linear and non-linear regression equations were used to fit between body weight and image characteristics [6]. The same technique was applied by Mollah et al. [7], who compared multivariate linear regression, Artificial Neural Networks (ANN), and Bayesian Artificial Neural Networks for weight estimation of the broiler flock aged from 24 to 34 days, with positive performance achieved. Amraei et al. [9,10] tried three models: support vector regression (SVR) [10], ANN with a (1-15-1) topology [9], and Transform Function [11], obtaining desirable results on fitting the body weight of broiler chickens aged from 1 to 42 days, with the best R^2^ being 0.98. Moreover, Wang et al. [12] applied ANN with a (1-19-1) topology trained by a backpropagation algorithm (BP) to estimate the body weight of broiler chickens aged from 3 to 30 days and gained excellent results, with the R^2^ being 0.9942 and RMSE being 0.048. However, the above-mentioned studies have mainly focused on the measurement of the body weight of young broilers. The body weight measurement of broilers older than 42 days has not been addressed, let alone of those which are raised for 60 days, 90 days, or even 120 days. The results of existing research indicate that when estimating the body weight of broiler chickens, the older the poultry become, the more errors arise in weight estimation because the chicken flocks grow more unevenly and the broiler house tends to be more crowded [8,9,10,11,12].

To tackle the above problems, we propose a framework based on multi-feature fusion and a gradient boosting decision tree, which is referred to as MFF-GBDT, for automatically and accurately estimating the body weight of broiler chickens with great day age. In comparison, existing approaches are not so friendly to the body weight estimation of older broiler chickens as they require some level of manual operation in image and feature selection, image segmentation, and morphometric measurement extraction.

The proposed framework is composed of three modules: an instance segmentation module, a feature fusion module, and a weight estimation module. In the first module, the depth image of a single broiler or multiple broilers will be used as input of the Mask R-CNN [14] to segment each broiler instance accurately and efficiently. Once the broiler instance is segmented by the instance segmentation module, it is extracted and fused with artificial and learned features from the target image and depth distance image in the second module. The third module adopts the fused features and the gradient boosting decision tree (GBDT) algorithm to estimate the body weight of broiler chickens. After the model is trained, the weight can be automatically obtained by inputting the image data. Encouragingly, the proposed framework performed well in our dataset with 200 bantam chickens (63-day-old).

The main contributions of our work are as follows: (i) The proposed framework can automatically and accurately estimate the body weight of older broiler chickens, outperforming the advanced ANN method, with an MAE of 0.093034 kg and an R^2^ of 0.706968 achieved in the test set for 63-day-old bantam chickens. (ii) We extracted new artificial features and designed a method of fusing learned features and artificial features, which can effectively reduce the error of weight estimation. (iii) A depth image dataset was established for the weight estimation of broiler chickens with great day age. What is more, the improved method can be universally applied to the weight estimation of other poultry.

## 2. Materials and Methods

### 2.1. Image and Weight Data Collection

We used a “Intel RealSense D435i” depth camera to collect the raw data from 63-day-old broiler chickens in Feilei Village, Leizhu Town, Xinxing County, Guangdong Province, from 4 February to 6 February 2021. The image data of a single broiler were collected from 200 bantam broiler chickens, and the image data of multiple broilers were collected from 286 bantam broiler chickens. A total of 1303 pseudo-color images and depth distance images were collected. Pseudocolor RGB images were mapped from depth images using the JET color mapping function, one of the most commonly used color mapping algorithms, which has a high contrast to effectively highlight broilers in the image. Therefore, unlike some research which selects the optimal color space method for different segmentation problems [15,16], we used a simple RGB color space for research. The partial data of single broiler images and multiple broiler images are shown in Figure 1. Notably, all animal procedures in this present study were approved by the South China Agricultural University’s Experimental Animal Ethics Committee (Protocol #2020c052).

The image acquisition method of a single broiler is shown in Figure 2. First, we fixed a depth camera on the top of an electronic scale (about 1 m above the ground) to capture the depth images of broilers vertically, one chicken at a time. While weighing the broiler, 5 or 6 images for each chicken were acquired. The depth image had a resolution of 1280 × 720 pixels, and 1198 pseudo-color images and depth distance images were collected in total.

On the other hand, to collect the depth images of multiple broilers, we held a depth camera and walked randomly in the net-raised chicken house, which is a more complex environment. At a height of about 1 m, we took pictures from the top when 2 to 3 chickens were in the field of view. After the broilers were photographed, each chicken was weighed individually and its weight information was recorded. In this way, 105 pseudo-color images (depth images) and depth distance images (distance matrix saved in Z16 format) with complex background were collected. Weight data were necessary for training the model in the experiment, so the broilers were photographed and weighed in the training stage. After the model training was completed, only the images and videos of broiler chickens were needed in real application through a camera set in a fixed height.

The histogram of the weight frequency distribution of 200 broiler chickens in a single broiler image set is shown in Figure 3. The mean weight was 1.598750 kg; the standard deviation was 0.276945; the maximum weight was 2.35 kg; and the minimum weight was 0.87 kg. Due to individual differences, broilers of the same age varied in body weight. We found that 66.61% of broilers weighed between [1.4, 1.9] kg, 22.45% of them weighed less than 1.4 kg, and 10.93% of them weighed more than 1.9 kg. This requires the model to be accurate enough to predict different weight levels of broilers of the same age, which is more difficult compared with the weight estimation of broilers of different ages.

### 2.2. Overview of the Proposed Framework

In order to accurately estimate the body weight of broiler chickens with great day age, a framework was constructed based on multi-feature fusion and a gradient boosting decision tree (MFF-GBDT), as shown in Figure 4. This proposed framework comprises three modules: (1) instance segmentation module (the preprocessing stage): its main function is to segment the collected bird’s eye-depth images of a single broiler or multiple broilers via a pre-trained instance segmentation network; (2) feature fusion module (the second stage): each instance image from the first stage is input to automatically extract and fuse artificial together with learned features; (3) weight estimation module: the GBDT model is trained by the fused features to estimate the body weight of each broiler chicken.

### 2.3. Broiler Instance Segmentation Based on Mask R-CNN

For the purpose of obtaining the target image of a single broiler, we used the superior Mask R-CNN model introduced by [14] to perform instance segmentation on the collected a single broiler image and multiple broiler images. The instance segmentation model can solve the segmentation problem caused by images with multiple broilers and complex background. In addition, this model can well balance the segmentation effect and segmentation efficiency.

Mask R-CNN [14] is a high-speed, high-accuracy algorithm that combines the classic target detection algorithm, Faster R-CNN [17], and the classic semantic segmentation algorithm, fully convolutional network (FCN) [18]. Faster R-CNN can quickly and accurately complete the function of target detection, while FCN can precisely perform the function of semantic segmentation. The Mask R-CNN structure diagram is shown in Figure 5, including the convolutional neural network (CNN), RPN (region proposal network), RoIAlign, class regression branch, bounding box (bbox) branch, and FCN mask branch. First, the convolution operation is performed on the depth image of broiler chickens to extract and fuse features in the CNN module. Within this framework, Resnet50 [19] and the feature pyramid network (FPN) are specifically used in the CNN module. Because Resnet50 is faster than Resnet101 and the semantic information of depth images is not complicated, there is no need to use a deeper feature extraction network. The RPN module is used to propose the candidate bbox of broilers, which can extract the candidate bbox with great efficiency based on anchors of different sizes and scales. Furthermore, for the pixel deviation problem of RoI pooling in Faster R-CNN, Mask R-CNN provides a corresponding RoIAlign strategy which can gain fixed-size feature maps of the bbox in various sizes without losing information. The class and bbox branches with multiple linear layers are harnessed to predict the classification and position deviation in the box. Finally, the mask branch composed of all convolutional layers can make accurate pixel mask predictions for each instance object.

Specifically, we used the Mask R-CNN model and interface implemented in the torchvision 0.8.1 library provided by pytorch to conduct a series of experiments. The model was pre-trained on the MSCOCO [20] dataset for transfer learning. More experimental settings are described in detail in Section 3.1. The segmentation model trained on the mixed images of a single broiler and multiple broilers was used to segment all the target chicken images in the image dataset with a single broiler. A total of 1198 target images were obtained as the input for the next stage, namely the feature extraction module.

### 2.4. Extraction and Fusion of Multi-Feature

In this section, we propose a method which combines artificial features with learned features, with the former based on manual design and extraction of 2D and 3D features, and the latter on convolutional extraction. With regard to the method of estimating the broiler weight based on image features [8,9,10,11,21], extracting features from broiler images is crucial as the quality of the extracted features will directly affect the accuracy of weight estimation. With the features appropriately selected, even if the hyperparameters are not optimal, the performance of the model is still impressive, so there is no need to spend too much time in finding the “best” parameters. This has greatly reduced the complexity of the model and made it simpler.

In the past, computer vision-based estimation of broiler weight used artificial features only. At present, we propose to utilise the learned features extracted from convolutional neural networks (CNN) which have performed strongly in the field of computer vision, such as object detection, image recognition, classification among others [22,23,24,25,26,27,28]. Following years of development and verification, CNN has showcased irrefutable superior performance in feature extraction. With richer features extracted by CNN, objects can be depicted in more detail. However, even if artificial features are frequently designed and desirable results have been attained in weight estimation, these features are not as comprehensive as they appear to be. After all, they are designed and extracted manually, and are constrained by a specialized dataset. Therefore, in order to balance the performance and universality in estimating broiler weight, we integrated learned features extracted by CNN with traditional artificial features designed by hand to mutually enhance functionality.

We adopted two methods to enrich the features used in weight estimation. One method extracted 16 new artificial features, plus 9 existing features and obtained a total of 25 artificial features. The other method used a combination of 2048 learned features extracted by CNN. The 25 artificial features were extracted by using image processing library functions of OpenCV in python language, while the learned features were extracted by Resnet50 [19] pre-trained with broiler weight regression, and 2048-dimensional feature vectors were extracted for each instance. After concatenating the learned feature and the artificial feature, a 2073-dimensional fused feature was obtained, as shown in Figure 6. In practice, we acquired the fused features on a set of instance images with size of batch-size (B) simultaneously.

#### 2.4.1. Feature Synthesis and Construction Based on 2D and 3D Information

Regarding the design and selection of artificial features, some existing features are inherited, such as projected area, contour perimeter, eccentricity, major axis length, minor axis length, convex hull area, width of the broiler, radius of largest inscribed circle, approximate volume, and height of the broiler. Nevertheless, some of the existing features have a high degree of similarity and, consequently, are computationally complex or difficult to extract. For instance, the back width is similar to the width of the broiler; convex volume is similar to volume; and surface area and convex surface area are similar to the projected area. Thus, these features will be removed or replaced in this study. Moreover, as mentioned previously, extracting features from the target broiler image with the head image removed causes information loss. Therefore, after acquiring the target image, we directly extracted features from the corresponding mask image and depth distance image without removing the head image.

In order to address the problem of insufficient reliable features, we extracted six new 2D geometric features based on the target image obtained by segmentation: the area and perimeter of the approximate contour, the perimeter of the convex hull, the maximum convexity defect which is the local maximum deviation of hull from contour, the approximate distance from hull to the red point in Figure 7, the sum of convexity defects, and the diameter of a circle equaled to the contour area. These features can describe broilers in different poses to a certain extent. Through feature synthesis and construction methods, two new 2D features were added, including the ratio of contour area to boundary rectangle area and the ratio of contour area to convex hull area. From the collected depth distance images, we extracted six new 3D statistical features that are simple and easy to obtain: maximum depth, minimum depth, average depth, depth range, depth standard deviation, and depth sum. With the aid of a feature construction method, two 3D features were added: one is the distance between the minimum depth and the average depth, and the other is the distance between the maximum depth and the average depth. The extracted features, both old and new, totalled 25. Table 1 presents the detailed features and identifiers. Among them, features 9 to 16 and features 18 to 25 are newly introduced. Features (1–16) represent 2D characteristics extracted from the target image, while features (17–25) correspond to 3D attributes extracted from the depth distance image. In addition, we also designed experiments to verify the effectiveness and importance of the proposed features. For the details of the experiment, see Section 3.2.1.

The boundary rectangular box, minimum fitting rectangular box, fitted ellipse, convex hull, approximate outline, and farthest point from hull are shown in Figure 7. Specifically, the yellow box is the boundary rectangular box; the white box is the minimum fitting rectangular box; the red ellipse is the fitted ellipse; the green line is the convex hull; the cyan line is the ap-proximate outline; and the red points in contour are the farthest point from hull. The projected area (Area) is calculated by Green’s Formula (1), where *L* is the closed contour curve, and *D* is the closed area enclosed by *L*. The approximate contour is calculated by the Douglas–Peucker algorithm with the epsilon parameter equaled to 0.01 times of the contour length.
(1)$$\begin{array}{l} \iint_{D}(\frac{\partial Q}{\partial x}-\frac{\partial P}{\partial y})dxdy=\oint_{L^{+}}Pdx+Qdy\\ Area=\iint_{D}dxdy=\oint_{L^{+}}xdy\;,\mathrm{when}\;P=0,Q=x \end{array}$$

The sum of the approximate convexity defects is formulated as
(2)$$\sum_{i=1}^{n}\frac{|A_{i}x_{i}+B_{i}y_{i}+C_{i}|}{\sqrt{A_{i}{}^{2}+B_{i}{}^{2}}}$$

The maximum approximate convexity defect is formulated as
(3)$$\max\left\{\frac{|A_{i}x_{i}+B_{i}y_{i}+C_{i}|}{\sqrt{A_{i}{}^{2}+B_{i}{}^{2}}}\right\}$$
where *n* is the number of convexity defects, (*x_i_*, *y_i_*) is the coordinate of the *i*th farthest point (red point), and A*_i_x* + B*_i_y* + C is the approximate straight line (green line) of the *i*th convexity defect.

The 3D point cloud image rendered from the depth distance image is shown in Figure 8. Since the shooting height distance is mostly fixed, the value of each pixel in depth distance image represents the distance from the lens plane to the point. The generated valley-shaped depth distance image of chickens are shown in Figure 8 and Figure 9a. From the depth distance image, we extracted the approximate volume of chicken and depth statistical features, including maximum depth, minimum depth, average depth, depth range, depth standard deviation, and depth sum. The approximate volume in the 3D features was the volume of the curved cylinder subtracting the volume of the curved cylinder surrounded by the valley-shaped curved surface. The result obtained is only an approximate volume, as shown in the green part of Figure 9b. The bottom area of the curved cylinder is the projected area in 2D feature, and the highest value is the maximum value of the edge, which is approximately the distance from the camera to the ground. The volume calculation formula is as follows:(4)$$Area*\;\max\left\{D(x_{p},y_{p})\right\}-\sum_{p=1}^{m}D(x_{p},y_{p})$$
where *Area* is the projected area; *m* is the number of target pixels; *D* is the depth distance; and (*x_p_*_,_*y_p_*) represents the *p*th pixel.

#### 2.4.2. Learned Features Extracted with C-Resnet50

In the proposed framework, we used C-Resnet50 to extract learned features instead of traditional ResNet50, where the fully connected layer is deleted. In order to make the learned features extracted by Resnet50 more applicable for weight prediction tasks instead of conventional classification or detection tasks, we made a small modification in ResNet50 to construct C-Resnet50 for weight regression training.

The original ResNet50 includes a fully connected layer and 49 convolutional layers. At the primary layer, ResNet50 utilizes a convolutional layer with a 3 × 3 filter size with a stride of 2 and a max pooling layer with a 3 × 3 filter size with a stride of 2 for downsampling. And then the image data are continuously convoluted by residual blocks in the following 4 stages, which are generally composed of 1 × 1 convolutions, 3 × 3 convolutions, and 1 × 1 convolutions. The number of residual blocks in the 4 stages are {3, 4, 6, 3}, respectively. After passing through the adaptive average pooling and the flatten operation, the size of the image pixel matrix is changed to batch-size × 2048. We extended the original ResNet50 with a fully connected (FC) layer for body weight regression training. The structure of C-Resnet50 with the extended FC layer is shown in Table 2. Each conv layer is followed by a BN and a ReLU layer. Residual operations are performed in each block of each stage. Downsampling is performed by the convolution in parallel with the first convolution of stages 2 to 5 with a stride of 2. To be specific, in order to extract weight-related features, the last FC layer of Resnet50 was extended to 3 FC layers, whose structures are (2048-1024-512-1). After the training was completed, we retained the model parameters and used the backbone of C-Resnet50 to extract learned features.

### 2.5. Broiler Weight Estimation Based on Gradient Boosting Decision Tree

The 25 artificial features extracted from the target image and the 2048 learned features extracted by C-Resnet50 are numerical features. To efficiently use the features for quick and accurate weight estimation of broiler chickens, we used the gradient boosting decision tree (GBDT) for modeling. The weight estimation experiment of broilers adopted the LightGBM model [29] and the XGBoost model [30], which implement the GBDT algorithm for training and prediction. The specific design of the experiment is shown in Section 3.2.

LightGBM is a gradient boosting framework that uses a tree-based learning algorithm. It is designed to be distributed and efficient, with faster training speed, greater efficiency, lower memory usage, and higher accuracy, which supports parallel, distributed, and GPU learning and is capable of handling large-scale data. XGBoost, as an optimized distributed gradient boosting library designed to be highly efficient, flexible, and portable, implements machine learning algorithms under the gradient boosting framework, providing a parallel tree boosting (also known as GBDT, GBM) that solves many data science problems in a fast and precise way.

The GBDT is an iterative decision tree algorithm in which each tree is fitted with a negative gradient. Trained with the extracted features, the sum of the conclusions of all trees in the algorithm is exactly the body weight of broiler chickens. Feature data and weight data are the components of the training sample. The whole training set S = {X*_mn_*, Y*_m_*}, where X*_mn_* is the feature data and Y*_m_* is the weight data vector. *m* represents the number of samples and indicates the dimension of feature. In this study, *m* = 1198. When using only artificial features, the dimension of *n* is 25. When just using learned features, the dimension of *n* is 2048. With both artificial features and learned features utilized, the dimension of *n* is 2073. The training set *S* was used to construct the gradient boosting regression tree algorithm, with F_K_(x) containing K regression decision trees. The pseudo-code of GBDT training with fusion features for broiler weight estimation is shown in Algorithm 1. A more detailed discussion is included in Appendix B.
**Algorithm 1** GBDT training with fusion features for broiler weight estimation**INPUT:**  S = { X*_mn_*, Y*_m_* }: Training set, X*_mn_* is fusion features with 2073 dimension, Y*_m_*           is body weight     K: The number of regression decision trees     ρ: The learning rate**OUTPUT:**  F_K_(x): Trained gradient boosting regression tree1. Set squared loss function: L(Ym,f(Xmn))=12(Ym,f(Xmn))22. Initializ f0(X)=Y¯, where Y¯ is the average weight, and *m* represents the number of samples.3. for each tree *k* in K do:4.  Calculate the negative gradient according to the loss function.     Y→m=−[∂L(Ym,f(Xmn))∂f(Xmn)]f(X)=fk−1(X)=Ym−fk−1(Xmn)5.  Establish a binary regression decision tree with *J* leaf region Rjk.6.  Calculate the predicted weight Cjk of each Rjk.      Cjk=argminc∑Xmn∈RjkL(Ym,fk−1(Xmn)+c)=avgXmn∈RjkY→m7.  Obtain the prediction of the kth regression decision tree and update result.       fk(X)=fk−1(X)+ρ∑j=1JCjkI(X∈Rjk)8. end for

### 2.6. Experimental Settings and Evaluation Metrics

All experiments, including the following instance segmentation experiments and weight estimation experiments, were implemented on a laptop computer. The CPU configuration was an Intel (R) Core (TM) i7-10750H CPU @ 2.60 GHz, and a GPU GeForce GTX 1660 Ti was used.

In the instance segmentation experiment, two evaluation metrics were selected: the average precision (Average Precision, AP) with IoU thresholds of 0.5 and 0.75 and the average recall (Average Recall, AR) under 10 IoU thresholds (0.50:0.05:0.95). The annotation data required for training were manually annotated using OpenCV’s open source CVAT software [31]. Collectively, 100 images with a single broiler and 105 images with multiple broilers were annotated, respectively. Examples of the depth images, annotation images, target images, and target mask images are shown in Figure 10. All randomly shuffled depth images and target images were used for training and testing, with 70% of them as the training set and 30% of them as the test set. The network was trained for 50 epochs with an initial learning rate of 0.001, a batch-size of 1, and a SGD optimizer. See the source code for more detailed experimental settings: https://github.com/GoldfishFive/MFF-GBDT (accessed on 29 August 2023).

We trained the C-Resnet50 for 100 epochs with a batch-size of 8, a learning rate of 0.001, and a fixed momentum of 0.9 with SGD optimizer. All parameters of C-Resnet50 were randomly initialized and learned.

The results of the weight estimation experiment were measured by the mean absolute error (MAE), the mean square error (MSE), the root mean square error (RMSE), and the coefficient of determination (R^2^) evaluation metrics, which are formulated as follows.
(5)$$MAE=\frac{1}{n}\sum_{i=1}^{n}\left|y_{i}-{\hat{y}}_{i}\right|$$
(6)$$MSE=\frac{1}{n}\sum_{i=1}^{n}{\left(y_{i}-{\hat{y}}_{i}\right)}^{2}$$
(7)$$RMSE=\sqrt{\frac{1}{n}\sum_{i=1}^{n}{\left(y_{i}-{\hat{y}}_{i}\right)}^{2}}$$
(8)$$R^{2}=1-\sum_{i=1}^{n}\frac{{\left(y_{i}-{\hat{y}}_{i}\right)}^{2}}{{\left(y_{i}-{\bar{y}}_{i}\right)}^{2}}$$

Here, $y_{i}\;$ is the actual body weight value (in kg); ${\hat{y}}_{i}$ is the estimated value of the model; ${\bar{y}}_{i}$ is the average value of GT (in kg); and *n* is the number of samples.

## 3. Results

### 3.1. Results of Broiler Instance Segmentation

In order to verify the segmentation effect of the instance segmentation model Mask R-CNN on the data of images with single broiler and multiple broilers, respectively, three groups of depth images (single broiler, multiple broilers, both single broiler and multiple broilers) were used for training and testing. The number of depth images in each group was 100, 105, and 205, respectively.

Table 3 shows the segmentation results of Mask RCNN trained with different types of data in the test set, in which AP refers to “Average Precision”, AR refers to “Average Recall”, and AP@.5IoU refers to the AP at IoU = 0.5. Among the results of the three experiments, indicators of AP@.5IoU and AP@.75IoU were close or equal to 1. Even for AP@.95IoU, the indicator was 0.861 in the segmentation test of mixed images. The segmentation effect of single broiler images was optimal, followed by that of multi-broiler images and mixed images, indicating that the result of segmentation network was close to completely correct image target classification. The average recall rate (AR) was about 0.9, which signifies that the model can segment the desired target well. For the target to be segmented, the mask must have a higher degree of correspondence with the pixels of the actual target. In general, the research outcome shows that the instance segmentation network performs well in segmenting and generalizing the data of images with a single broiler or multiple broilers in a complex background. Figure 11 illustrates the qualitative results of broiler instance segmentation on the test split. Compared with the data instance segmentation of images with a single broiler, the efficiency of segmentation and extraction of the target image increases by N times for the data instance segmentation of multiple broilers, where N is the number of broilers in an image with multiple broilers.

### 3.2. Feature Analysis, Fusion and Weight Estimation

#### 3.2.1. Artificial Feature Analysis

To verify that the proposed features (feature numbers 9–16, 18–25 in Table 1) are beneficial for weight estimation, we performed multiple contrastive experiments. First, we used all the artificial features, including the existing and proposed features, and designed a contrastive experiment to select a more outstanding model between the GBDT model and the ANN model and, in terms of effects, a fair comparison between the proposed features and the existing features could be made. The results are displayed in Table 4, where ‘Normalization’ refers to the normalization of all features. ‘AF’ denotes all artificial features. ‘ANN_1’ represents an artificial neural network with a (25-19-1) topology; similarly, ‘ANN_2’ has a (25-64-32-1) topology, and ‘ANN_3’ features a (25-1024-512-1) topology. The optimal results are highlighted in bold.

Different architectures of ANN (ANN_1, ANN_2, ANN_3) are shown in Table 5. ANN_1 indicates artificial neural network used by [8] with a (25-19-1) topology, while ANN_2 has a (25-64-32-1) topology, and ANN_3 a (25-1024-512-1) topology. For ANN training, we employed a stochastic gradient descent optimizer for 200 epochs. A batch-size of 8, an initial learning rate of 0.001, a loss function of L1 loss, and a weight decay with 0.05 were used.

Two library functions (LGBM and XGB) were implemented the GBDT algorithm. We used LGBM python package with version 3.3.0 from the GitHub repository of Microsoft et al. [32] and the XGB package with version 1.3.3 from Dmlc et al. [33]. Both models kept identical hyperparameters in all experiments: for LGBM training, an n_estimators of 4000 and a learning rate of 0.1 were used. Other parameters were set as num_leaves = 15, max_depth = 5, min_child_samples = 15, min_child_weight = 0.01, subsample = 0.8, and colsample_bytree = 1. For the XGB training, we trained the models with an n_estimators of 2000 and a learning rate of 0.1. Other parameters were set as max_depth = 5, learning_rate = 0.1, objective = ‘reg:linear’, min_child_weight = 1, max_delta_step = 0, subsample = 0.8, colsample_bytree = 0.7, colsample_bylevel = 1, reg_alpha = 0, reg_lambda = 1, and scale_pos_weight = 1.

As shown in Table 4, when modeling all the artificial features, ANN_3 obtained an MAE of 0.139806 kg and an R^2^ of 0.295418, indicating that ANN_3 performed better than ANN_1 and ANN_2, because it obtained an MAE 0.002175 kg–0.004561 kg lower and an R^2^ 0.220583-0.280782 higher than those in ANN_1 and ANN_2. The result reveals that a wider and deeper ANN can better estimate body weight when using more features. However, the combination of AF and GBDT (LGBM/XGB) outperformed the combination of AF and ANN_3, with an MAE about 0.010 kg smaller and an R^2^ about +0.26 higher than the latter combination. It is illustrated that the machine learning GBDT algorithm is superior to the artificial neural network (ANN) in estimating the weight of broiler chickens of the same day age when artificial features are used. Moreover, compared with the approach of directly entering the extracted features into the model for training and learning, normalizing features in advance can achieve more satisfactory performance when using XGB for modeling, but the difference is insignificant. The test set with a broiler body weight ranging from 0.85 kg to 2.35 kg was divided into 5 groups, and the mean absolute error was calculated separately for visualization. The difference between each group was 0.3 kg, and the number of images in each group was 16, 45, 108, 61, and 10, respectively. The MAE results are shown in Figure 12. The MAE of the first weight class was around 0.2 kg, and the MAE of the medium weight class was around 0.13 kg, which suggests that the model estimated heavier broilers with better performance. The last weight class with a small number of samples for training had an MAE of around 0.3 kg in test set.

With the GBDT algorithm selected, we evaluated the performance of using different features to estimate the broiler weight. Table 6 lists the results of XGB and LGBM on the existing features, the proposed features, and the artificial features. The existing features comprise feature numbers 1 to 8 and 17, as listed in Table 1. The proposed features consist of feature numbers 9 to 16 and 18 to 25, also from Table 1. Lastly, the artificial features encompass feature numbers 1 to 25 from Table 1. The features proposed by us consistently bought +0.18–0.20 R^2^ gains over existing features, with a lower MAE (0.028619 kg–0.030131 kg). This proves that the proposed artificial features are more reliable and stable when estimating the broiler weight of broiler chickens of great day age. Moreover, a further improvement in performance can be observed when all features are used, providing an MAE of 0.129030 kg and an R^2^ of 0.555427. In comparison with the proposed features used alone, the use of proposed features achieved an R^2^ +0.015686 higher under XGB and +0.003315 higher under LGBM. The improved performance can be attributed to the construction and synthesis of new artificial features, which have provided a more detailed and richer appearance to features of the target broiler.

#### 3.2.2. Ablation Study of Multi-Feature

Table 7 shows the performance in test set of different models with learned features alone and with both artificial and learned features as input. According to the results, an MAE of 0.103644 kg could be achieved by employing the learned features and the artificial neural network ANN_3 to perform a regression prediction of weight. Meanwhile, the error margin was smaller (0.036162 kg, 0.103644 kg vs. 0.139806 kg), and the R^2^ was higher (0.433728, 0.729146 vs. 0.295418) compared with using artificial features modeling alone (see Table 4, AF + ANN 3). Moreover, it can be seen that using the ML method GBDT instead of ANN to model the extracted learned features by Resnet50 we could achieve an even smaller MAE (0.004937 kg–0.009492 kg). The results of XGB and LGBM were 0.013770 kg–0.014094 kg (MAE) smaller and 0.053760–0.054201 (R^2^) higher than ANN_3 when using both artificial and learned features. As shown in Figure 13, it can be observed that under the proposed MFF-GBDT framework, improved performance is achieved by combining different feature components with weight estimation for the broilers of great day age. Figure 13 depicts the MAE and R^2^ plots of our proposed framework with different feature combinations in the test set, using LGBM, XGB, and ANN_3. These feature combinations include existing features (EF), proposed features (PF), artificial features (AF), learned features (LF), and artificial and learned features (AF and LF).

## 4. Discussion

It is economically important to obtain the weight information of broilers in chicken farms, as weight is a crucial indicator of the broiler growth. Traditional manual weight estimation methods are not only time-consuming and labor-intensive but also have the potential to induce stress reactions in broiler chickens [2,3,4,5]. Furthermore, existing research has predominantly focused on weight measurements for young broiler chickens, with little attention given to broilers aged 42 days or older. Existing research findings suggest that as broiler chickens grow older, the weight estimation error increases. This is due to the less uniform growth of the chicken population and the increased crowding within broiler houses [8,9,10,11,12].

We address these challenges. Firstly, we have introduced the MFF-GBDT framework, which is capable of non-invasively, efficiently, and accurately estimating the weight of broiler chickens. Moreover, it performs exceptionally well with older broiler chickens, achieving a low MAE of 0.093034 kg and an R^2^ value of 0.706968. This framework is currently being implemented in practical application within the industry. Secondly, for high-value broiler chickens, the framework utilizes depth cameras for image capture, followed by a unique feature fusion method that integrates artificial 2D and depth 3D features. This integration effectively reduces weight estimation errors. Thirdly, we have made our code open-source and provided a depth image dataset that we created ourselves (https://github.com/GoldfishFive/MFF-GBDT (accessed on 29 August 2023)). This dataset is suitable for estimating the weight of older chickens. Additionally, our image capture devices can easily be transformed into portable handheld devices for practical applications. Furthermore, the improvements we made can be universally applied to weight estimation for various other poultry species.

One limitation of the present study is that our dataset has not included broiler chickens of significant age, such as 120-day-old broilers. Another limitation is that manual data collection is still required in the early stage for subsequent model training. In the future, we will consider creating depth image datasets with a variety of day age groups for broiler weight estimation. Meanwhile, due to the abnormal weight estimation caused by the posture of the broiler, an additional detection module can be considered in the preprocessing stage. In order to more accurately estimate the weight of broiler chickens of older day age, we will make an attempt to use multi-camera data for weight estimation in our future work.

Despite these limitations, future work will aim to address these issues and explore multi-camera data for more accurate weight estimation. The proposed method holds promise for wider applications in intelligent agriculture, extending to various poultry species beyond broilers.

## 5. Conclusions

In this paper, we present MFF-GBDT, a simple, effective, and accurate method for broiler weight estimation, which contains an instance segmentation module, a feature fusion module, and a weight estimation module. The instance segmentation module can aptly address the problem concerning depth images with complex background and accurate segmentation of multiple broilers, leading to both high efficiency and ameliorated performance. In the feature fusion module, some new and effective artificial features are proposed. A method of fusing artificial features and learned features is designed, which can effectively reduce the error in estimating the weight of broiler chickens of great day age. MFF-GBDT employs gradient boosting decision tree (GBDT)-integrated fusion features for broiler weight estimation, achieving an MAE of 0.093034 kg, an MSE of 0.019608, an RMSE of 0.140027, and an R^2^ of 0.706968 in the test set of 63-day-old bantam chickens, which were 0.046772 kg smaller (MAE) and 0.41155 higher (R^2^) than the advanced approach ANN which did not use our proposed new artificial features and learned features. Hopefully our method can serve as a solid baseline for further research of poultry weight estimation.

We are willing to promote our method for a wider application (e.g., intelligent breeding, breeding management, and health assessment) in the field of intelligent agriculture. Regarding the potential application of this method in the intelligent poultry industry, our method can be readily transferred to the body weight measurement of similar poultry such as ducks, geese, turkeys, quails, and other livestock.

## Figures and Tables

**Figure 1 animals-13-03721-f001:**
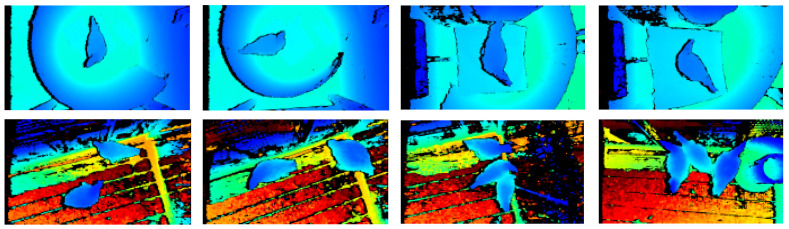
The partial data of a single broiler image and multiple broilers images.

**Figure 2 animals-13-03721-f002:**
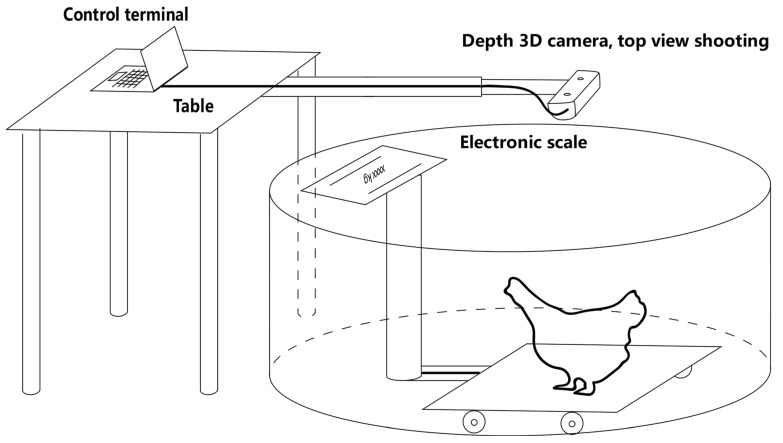
Schematic diagram of a single broiler image shooting.

**Figure 3 animals-13-03721-f003:**
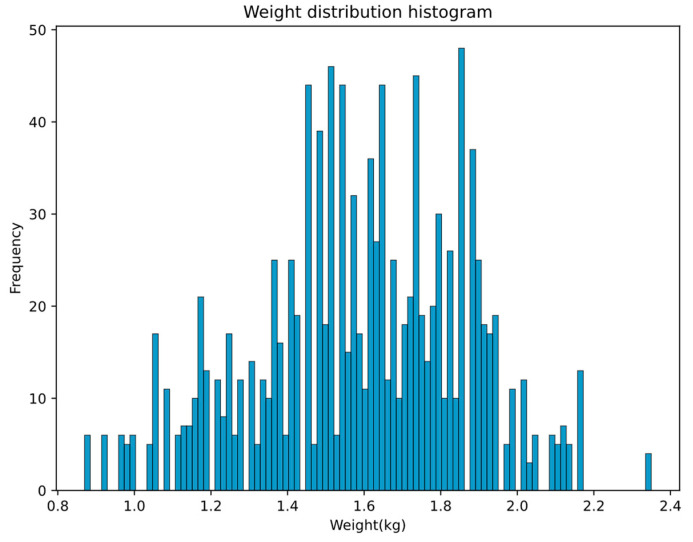
Body weight distribution histogram of 200 broiler chickens.

**Figure 4 animals-13-03721-f004:**
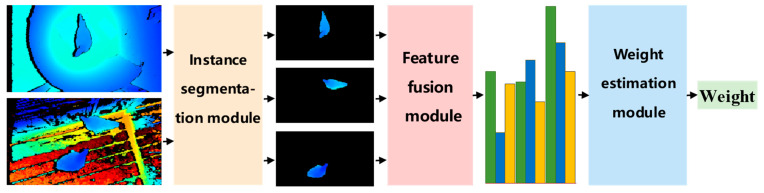
Flow chart of the weight estimation method of broiler chickens.

**Figure 5 animals-13-03721-f005:**
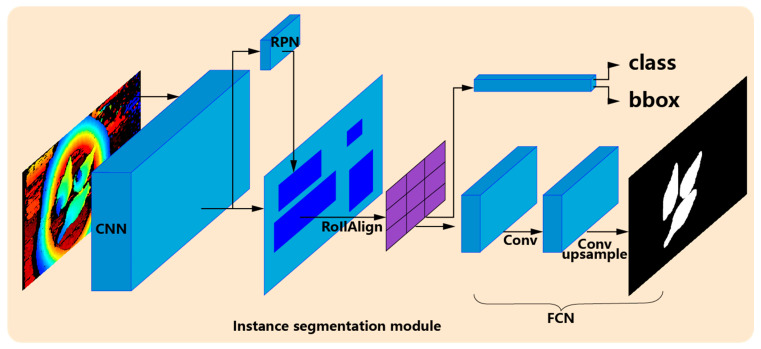
Instance segmentation module and architecture of Mask R-CNN.

**Figure 6 animals-13-03721-f006:**
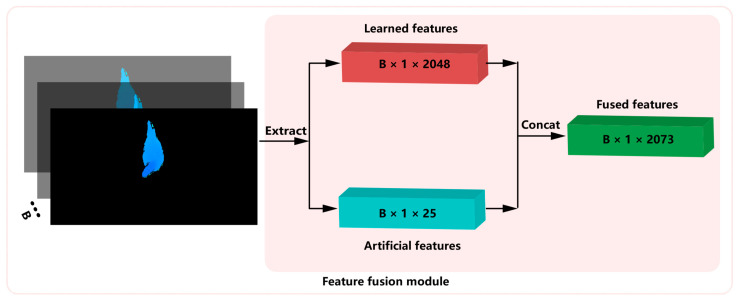
Fusion of artificial and learned features.

**Figure 7 animals-13-03721-f007:**
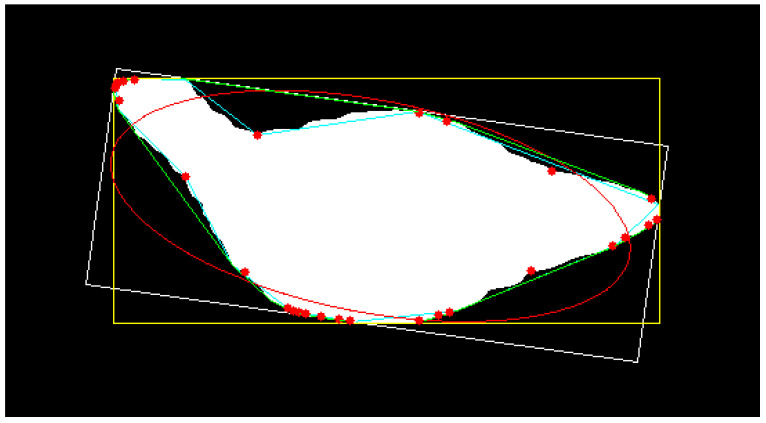
Visualizing geometric properties.

**Figure 8 animals-13-03721-f008:**
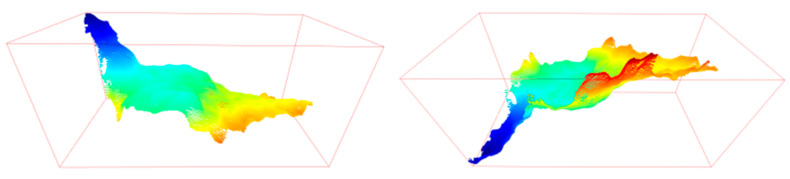
3D point cloud map.

**Figure 9 animals-13-03721-f009:**
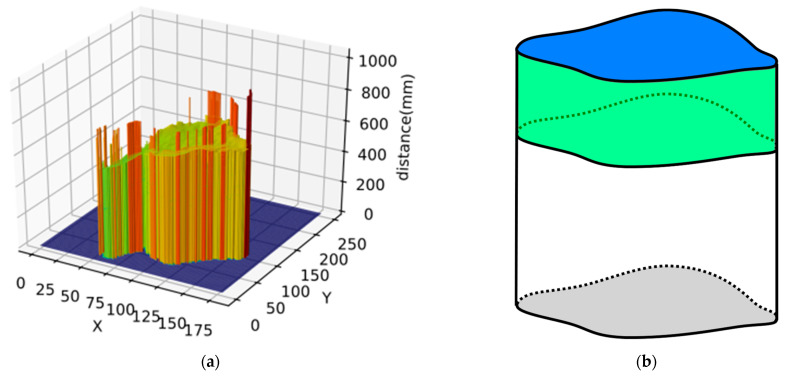
(**a**) 3D distance bar. (**b**) Schematic diagram of volume.

**Figure 10 animals-13-03721-f010:**
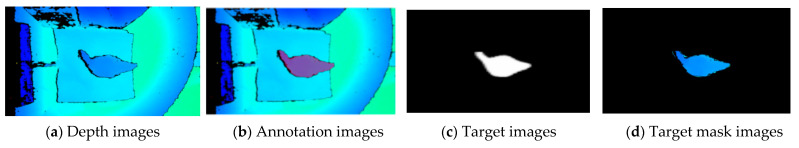
Different types of images.

**Figure 11 animals-13-03721-f011:**
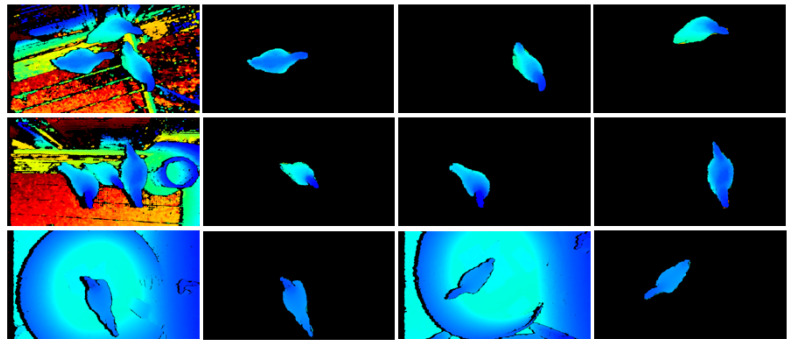
Qualitative results of broiler instance segmentation on test split. Two examples of multiple broilers (row 1 and row 2) with depth images (col.1) and segmentation results (col.2–col.4). Two examples of a single broiler (row 3) with depth images (col.1 and col.3) and segmentation results (col.2 and col.4).

**Figure 12 animals-13-03721-f012:**
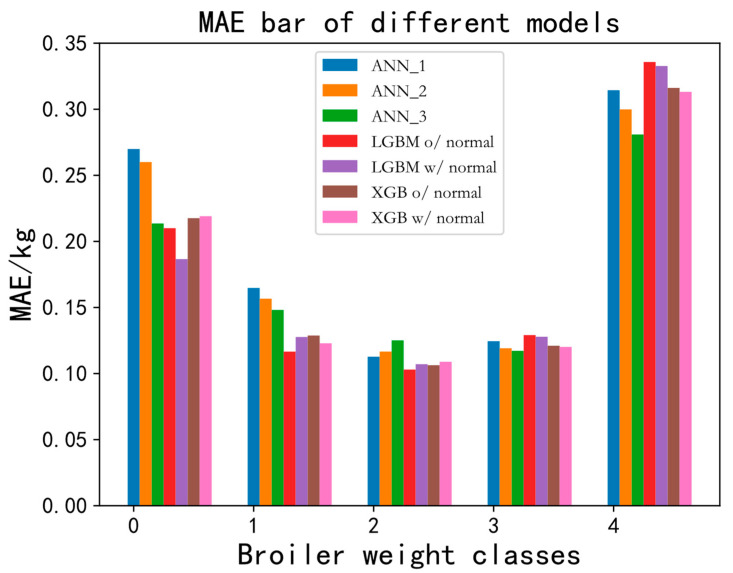
Mean absolute error of estimated body weight in different models for broiler chickens in 5 weight classes in the test set.

**Figure 13 animals-13-03721-f013:**
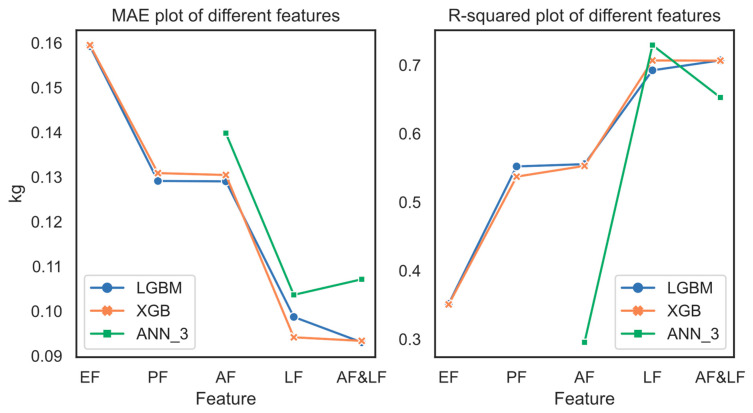
Impact of feature combinations on performance.

**Table 1 animals-13-03721-t001:** The extracted artificial features and identifiers.

Feature	
1. Projected area	14. Maximum convexity defect
2. Contour perimeter	15. Sum of the convexity defects
3. Width of the broiler	16. Diameter of a circle equaled to the contour area
4. Height of the broiler	17. Approximate volume
5. Convex hull area	18. Maximum depth
6. Minor axis length	19. Minimum depth
7. Major axis length	20. Average depth
8. Eccentricity	21. Depth range
9. Convex hull perimeter	22. Depth standard deviation
10. Approximate contour area	23. Depth sum
11. Approximate contour perimeter	24. Distance between the minimum depth and theaverage depth
12. Ratio of contour area to boundary rectangular area	25. Distance between the maximum depth and the average depth
13. Ratio of contour area to convex hull area	

**Table 2 animals-13-03721-t002:** Structure of C-Resnet50 with extended FC layer.

Stages 0–1		Stages 2–3		Stages 4–5	
Conv(7 × 7,64, stride 2) Max pool(stride 2)	×1	Conv(1 × 1,128) Conv(3 × 3,128) Conv(1 × 1,512)	×4	Conv(1 × 1,512) Conv(3 × 3,512) Conv(1 × 1,2048)	×3
Conv(1 × 1,64) Conv(3 × 3,64) Conv(1 × 1,256)	×3	Conv(1 × 1,256) Conv(3 × 3,256) Conv(1 × 1,1024)	×6	Average pool, flatten FC_0 (1024) FC_1(512) FC_2(1)	×1

**Table 3 animals-13-03721-t003:** The segmentation performance of different types of data on Mask RCNN in test set.

Training Data	AP@.5IoU	AP@.75IoU	AP@.95IoU	AR
Images with a single broiler	1.000	1.000	0.894	0.900
Images with multiple broilers	1.000	0.993	0.872	0.881
Mixed images	0.996	0.989	0.861	0.872

**Table 4 animals-13-03721-t004:** Results acquired from all artificial features in different models in the test set, and the optimal results are highlighted in bold.

Feature + Model	Normalization	MAE ↓	MSE ↓	RMSE ↓	R^2^ ↑
AF + ANN_1	√	0.144367	0.037025	0.192419	0.014636
AF + ANN_2	√	0.141981	0.036193	0.190245	0.074835
AF + ANN_3	√	0.139806	0.034603	0.186019	0.295418
AF + LGBM		**0.129030**	0.029747	0.172475	0.555427
AF + LGBM	√	0.130942	0.030435	0.174458	0.545145
AF + XGB		0.130452	0.029916	0.172962	0.552913
AF + XGB	√	0.130262	**0.029699**	**0.172333**	**0.556155**

**Table 5 animals-13-03721-t005:** Architectures of ANN_1, ANN_2, and ANN_3.

ANN_1	ANN_2	ANN_3
FC_layer_0 (input,19)	FC_layer_0 (input, 64)	FC_layer_0 (input, 1024)
ReLU_0	ReLU_0	LayerNorm (1024)
FC_layer_1 (19, 1)	FC_layer_1 (64, 32)	ReLU_0
	ReLU_1	FC_layer_1 (1024, 512)
	FC_layer_2 (32, 1)	LayerNorm (512)
		ReLU_1
		FC_layer_2 (512, 1)

**Table 6 animals-13-03721-t006:** Comparison of different features for body weight estimation in the test set, and the optimal results are highlighted in bold.

Model	Feature	MAE ↓	MSE ↓	RMSE ↓	R^2^ ↑
XGB	Existing features	0.159492	0.043444	0.208432	0.350732
	Proposed features	0.130873	0.030965	0.175970	0.537227
	Artificial features	0.130452	0.029916	0.172962	0.552913
LGBM	Existing features	0.159246	0.043391	0.208305	0.351526
	Proposed features	0.129115	0.029969	0.173117	0.552112
	Artificial features	**0.129030**	**0.029747**	**0.172475**	**0.555427**

**Table 7 animals-13-03721-t007:** Performance of different models in the test set with learned features alone and with both artificial and learned features as input, and the optimal results are highlighted in bold.

Feature	Model	MAE ↓	MSE ↓	RMSE ↓	R^2^ ↑
Learned features	ANN_3	0.103644	0.020305	0.142496	**0.729146**
	XGB (Ours)	0.094152	0.019622	0.140077	0.706757
	LGBM (Ours)	0.098707	0.020585	0.143475	0.692357
Artificial and learned features (Ours)	ANN_3	0.107128	0.023234	0.152428	0.652767
	XGB (Ours)	0.093358	0.019637	0.140132	0.706527
	LGBM (Ours)	**0.093034**	**0.019608**	**0.140027**	0.706968

## Data Availability

See the source code for more detailed experimental settings: https://github.com/GoldfishFive/MFF-GBDT (accessed on 29 August 2023).

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
