# Peer review of "An Improved Method for Broiler Weight Estimation Integrating Multi-Feature with Gradient Boosting Decision Tree"

_animals, 2023, doi:10.3390/ani13233721_

Round 1

Reviewer 1 Report

Comments and Suggestions for Authors

The article is devoted to solving the applied problem of image classification. The topic of the article is relevant. The structure of the article does not correspond to that accepted in MDPI for research articles (Introduction (including analysis of analogues), Models and methods, Results, Discussion, Conclusions). The level of English is acceptable. The article is easy to read. The figures in the article are of poor quality and too small. The article cites 39 sources, many of which are not relevant.

The following comments and recommendations can be formulated regarding the material of the article:

1. The title of the article contains the term “decision tree”, but as such this term appears mainly in section 2.6. The question arises: how do authors classify images? Why is “decision tree” not implemented as such?

2. Here the authors will of course tell me that “decision tree” is GBDT, and enough has been written about this GBDT in the article. BUT not everything is so simple. The fact is that the authors do not use GBDT, but someone else’s (framework). There is nothing wrong with this, it’s just that in this case there is no scientific novelty. So what is the scientific novelty of your research, dear authors?

3. You cannot talk about the use of machine learning or artificial intelligence in isolation from data. And in the article, the authors specifically mention “our data.” The problem is that there is no link through which I could familiarize myself with this dataset. In addition, the authors can prove that their dataset is representative for training, for example, the mentioned R-CNN.

4. By the way, when mentioning R-CNN, I will not forget to say that this type of neural networks suffers from myopia. These neural networks are good at recognizing small details in images (which is why they are often used to detect contours), but they are bad at perceiving the picture as a whole. Transformers do the latter well. So complex transformer+CNN neural networks are now popular. Why don't authors use this?

5. The authors solve the classification problem and evaluate the results in metrics (5)-(8). This is a good, proven metric, but for a regression problem. Where are the errors of the first and second types, dear authors?

Author Response

Comments from Reviewer 1

1.The title of the article contains the term decision tree, but as such this term appears mainly in section 2.6. The question arises: how do authors classify images? Why is decision tree not implemented as such?

Thank you for your comment. Gradient boosting algorithm is a Machine learning technology for regression, classification and ranking tasks, and the learner based on Gradient Boosting algorithm is called GBM(Gradient Boosting Machine). In theory, GBM can choose a variety of different learning algorithms as base learners. When the basis learner we use is a Decision Tree, then the Gradient Boosting algorithm is the gradient Boosting decision Tree GBDT(Gradient Boosting Decision Tree), also known as MART (Multiple Additive Regression Tree). It is an iterative decision tree algorithm, which consists of multiple decision trees, and the conclusions of all trees add up to get the predicted weight. I sincerely hope that the above explanation can be helpful.

2.Here the authors will of course tell me that decision tree is GBDT, and enough has been written about this GBDT in the article. BUT not everything is so simple. The fact is that the authors do not use GBDT, but someone elses (framework). There is nothing wrong with this, its just that in this case there is no scientific novelty. So what is the scientific novelty of your research, dear authors?

Thank you for your comment. In this paper, we use XGBoost and XGB to predict body weight, which are efficient systematic implementations of the Gradient Boosting algorithm. Our proposed framework consists of a series of operations: first, taking depth images, then extracting and fusing artificial and learned features, and finally using GBDT to predict body weight. Among them, extracting and fusing multi-source features is the core of our method.

3.You cannot talk about the use of machine learning or artificial intelligence in isolation from data. And in the article, the authors specifically mention our data. The problem is that there is no link through which I could familiarize myself with this dataset. In addition, the authors can prove that their dataset is representative for training, for example, the mentioned R-CNN.

Thank you for your comment. We have released the training and testing code. Data descriptions and downloads can be found in the link (Lines659-660). There are two kinds of data, one for training Mask R-CNN and the other for extracting artificial and learned features and training the GBDT algorithm.

4.By the way, when mentioning R-CNN, I will not forget to say that this type of neural networks suffers from myopia. These neural networks are good at recognizing small details in images (which is why they are often used to detect contours), but they are bad at perceiving the picture as a whole. Transformers do the latter well. So complex transformer+CNN neural networks are now popular. Why don't authors use this?

 Thank you for your comment. To segment broiler instances, we use an instance segmentation model, Mask R-CNN, instead of the target detection model, R-CNN. Since the broiler segmentation task is not difficult, using the simple convolutional model Mask R-CNN is more efficient than the complex Transformer+CNN model. It is worth mentioning that we extracted the learned features using a simple convolutional model called ResNet50, which is widely used in the community. Many thanks again

5.The authors solve the classification problem and evaluate the results in metrics (5)-(8). This is a good, proven metric, but for a regression problem. Where are the errors of the first and second types, dear authors?

Thank you for your comment. Our task was to predict the weight of broilers rather than the weight classification. MAE and MSE are important measures of regression error (between true weight and predicted weight of broiler).

Reviewer 2 Report

Comments and Suggestions for Authors

1. Authors summarize clearly what the difference/superiority of this work compared with the existing methods, especially in abstract/introduction/conclusion sections. For example, in abstract, it is difficult to understand the difference/superiority of this work with these key words (Mask RCNN, depth camera, feature fusion, decision tree).

2. In section 3, authors need to specify the execution times.

3. Reference list is not complete. Pls include other weight estimation papers for pig/cow/....

Author Response

Comments from Reviewer 2

1.Authors summarize clearly what the difference/superiority of this work compared with the existing methods, especially in abstract/introduction/conclusion sections.  For example, in abstract, it is difficult to understand the difference/superiority of this work with these key words (Mask RCNN, depth camera, feature fusion, decision tree).

Thank you for your comment. This work presents a framework for estimating broiler body weight based on computer vision. Mask R-CNN is used to segment broilers in depth images. Fusion features (artificial and learned features) were used to predict broiler body weight. The prediction model is GBDT algorithm based on decision tree as meta-learner. Hoping the above explanation is helpful to you.

2.In section 3, authors need to specify the execution times.

Thank you for your comment. We have rewritten this sentence to clarify it. It takes about 0.1732s to process an image (0.1732s=0.1550s+0.0100s+0.008s)

  • Average time to segment single broiler images: 0.1550s
  • Average feature extraction time: 0.0100s
  • Mean time to predict weight with GBDT: 0.0082s

The above statistics were based on a laptop computer with a configuration of Intel (R) Core (TM) i7-10750H CPU @ 2.60GHz, and GeForce GTX 1660 Ti GPU. We can find that the main time using is in segmentation module with the deep learning model Mask R-CNN.

3.Reference list is not complete.  Pls include other weight estimation papers for pig/cow/....

Thank you for your comment. We added and quoted the following 5 articles on weight estimation of pigs, cattle, and sheep and 2 articles on image processing:

Dohmen, R.; Catal, C. & Liu, Q. Computer vision-based weight estimation of livestock: a systematic literature review. New Zealand Journal of Agricultural Research 2022, 65, 227-247.

Ruchay, A.N.; Kober, V.; Dorofeev, K.A.; Kolpakov, V.; Dzhulamanov, K.M.; Kalschikov, V.V.; Guo, H. Comparative analysis of machine learning algorithms for predicting live weight of Hereford cows. Computers and Electronics in Agriculture 2022, 195, 106837.

Gjergji, M.; et al. Deep Learning Techniques for Beef Cattle Body Weight Prediction. In Proceedings of the International Joint Conference on Neural Networks (IJCNN), Glasgow, UK, 19-24 July 2020; pp. 1-8.

Kwon, K., Park, A., Lee, H., & Mun, D. Deep learning-based weight estimation using a fast-reconstructed mesh model from the point cloud of a pig. Computers and Electronics in Agriculture 2023, 210, 107903.

Mahmud, M. S., Zahid, A., Das, A. K., Muzammil, M., & Khan, M. U. A systematic literature review on deep learning applications for precision cattle farming. Computers and Electronics in Agriculture 2021, 187, 106313.

García-Mateos, G., Hernández-Hernández, J. L., Escarabajal-Henarejos, D., Jaén-Terrones, S., & Molina-Martínez, J. M. Study and comparison of color models for automatic image analysis in irrigation management applications. Agricultural water management 2015, 151, 158-166. 

Hernández-Hernández, J. L., García-Mateos, G., González-Esquiva, J. M., Escarabajal-Henarejos, D., Ruiz-Canales, A., & Molina-Martínez, J. M. Optimal color space selection method for plant/soil segmentation in agriculture. Computers and Electronics in Agriculture 2016, 122, 124-132. 

Reviewer 3 Report

Comments and Suggestions for Authors

An improved method for broiler weight estimation integrating multi-feature with gradient boosting decision tree 

1. Very interesting research entitled “An improved method for broiler weight estimation integrating multi-feature with gradient boosting decision tree”.

2. The structure of the article complies with the format of the MDPI-animals journal. 

3. The references in the article must be in progressive form. See the following figures in the attached file.

4. The figures should not be on a single line. Cases: figures 2-3 and 8-9. Correct. (see attached file)

5. The title of the figures (7 and 13) must be short. The previous paragraph should explain the figure.

6. The title of the tables (1, 2, 3, 4 and 6) should be short. In the previous paragraph, the table must be explained.

7. Table 1 has 2 columns with the same title, the text must wrap to the left, there is information in bold in the table (not necessary). Correct the table.

8. The section “2.1. Image and weight data collection”, pseudocolor RGB photos were taken.

·       Was any preprocessing done to crop the image and to correct: noise, high light, low light, shadows, etc.?

·       Have you tried converting the image to another color space (XYZ, L*a*b*, L*u*v*, HSV, HLS, YCrCb, YUV, I1I2I3, TSL, etc.) to seek more precision in the results?

9. Since the article is about image processing, I suggest you check out the following articles:

·        García-Mateos, G., Hernández-Hernández, J. L., Escarabajal-Henarejos, D., Jaén-Terrones, S., & Molina-Martínez, J. M. (2015). Study and comparison of color models for automatic image analysis in irrigation management applications. Agricultural water management, 151, 158-166.  https://doi.org/10.1016/j.agwat.2014.08.010

·        Hernández-Hernández, J. L., García-Mateos, G., González-Esquiva, J. M., Escarabajal-Henarejos, D., Ruiz-Canales, A., & Molina-Martínez, J. M. (2016). Optimal color space selection method for plant/soil segmentation in agriculture. Computers and Electronics in Agriculture, 122, 124-132.  https://doi.org/10.1016/j.compag.2016.01.020

10. I suggest developing a GBDT algorithm indicated in the section “3.2. Feature analysis, fusion and weight estimation”.  I suggest that the algorithms in this article use the following format: (see attached file).

11. Very good bibliography. I hope you can consult more bibliography.

Note: The points mentioned in the review are recommendations from another perspective that gives the authors the opportunity to improve their article. The observations are formulated to propose a different point of view and it is important that the authors address and implement them in their entirety.

Author Response

Comments from Reviewer 3

1.Very interesting research entitled An improved method for broiler weight estimation integrating multi-feature with gradient boosting decision tree.

Thank you very much for your interest in our work. We really hope that our work will shed some light on poultry weight.

2.The structure of the article complies with the format of the MDPI-animals journal. 

Thank you for your comment. We have added acknowledgements, Appendix 1 and Appendix 2 according to the suggested structure.

3.The references in the article must be in progressive form. See the following figures in the attached file.

Thank you for your comment. We made the modification and cited the references in a progressive form according to the suggestion.

4.The figures should not be on a single line. Cases: figures 2-3 and 8-9. Correct. (see attached file)

Thank you for your comment. We made a change and only put one image per row as suggested.

5.The title of the figures (7 and 13) must be short. The previous paragraph should explain the figure.

Thank you for your comment. We have simplified the title of the image. The corresponding explanation is added to the article(Figure 7: L342-346)(Figure 13: L609-615).

6.The title of the tables (1, 2, 3, 4 and 6) should be short. In the previous paragraph, the table must be explained.

Thank you for your comment. We have modified the title of the table and made necessary explanations for the corresponding positions in the article.(Table 1: L330-335), (Table 2: L391-397), (Table 3: L471-473), (Table 4: L510-514), (Table 6: L578-582)

  1. Table 1 has 2 columns with the same title, the text must wrap to the left, there is information in bold in the table (not necessary). Correct the table.

Thank you for your comment. As suggested, we removed the redundant title, left justified the text, and removed the bold formatting. Thank you for your valuable advice.

8.The section “2.1. Image and weight data collection”, pseudocolor RGB photos were taken.

Was any preprocessing done to crop the image and to correct: noise, high light, low light, shadows, etc.?

Have you tried converting the image to another color space (XYZ, L*a*b*, L*u*v*, HSV, HLS, YCrCb, YUV, I1I2I3, TSL, etc.) to seek more precision in the results?

Thank you for your comment. First of all, depth cameras work well in both bright and low light environments. Secondly, the segmentation model is robust enough, and the noise does not affect the extraction of artificial features, so we do not do any preprocessing.

Pseudocolor RGB images are mapped from depth images using the JET color mapping function, one of the most commonly used color mapping algorithms, which has a high contrast to effectively highlight broilers in the image. So we use a simple RGB color space for research.

9.Since the article is about image processing, I suggest you check out the following articles:

  • García-Mateos, G., Hernández-Hernández, J. L., Escarabajal-Henarejos, D., Jaén-Terrones, S., & Molina-Martínez, J. M. (2015). Study and comparison of color models for automatic image analysis in irrigation management applications. Agricultural water management, 151, 158-166.  https://doi.org/10.1016/j.agwat.2014.08.010
  • Hernández-Hernández, J. L., García-Mateos, G., González-Esquiva, J. M., Escarabajal-Henarejos, D., Ruiz-Canales, A., & Molina-Martínez, J. M. (2016). Optimal color space selection method for plant/soil segmentation in agriculture. Computers and Electronics in Agriculture, 122, 124-132.  https://doi.org/10.1016/j.compag.2016.01.020

Thank you for your comment. We have studied these references and cited them in our article. (L168-169) Thank you very much for your suggestions and we would be more than happy to try these methods in future studies.

  1. I suggest developing a GBDT algorithm indicated in the section 2. Feature analysis, fusion and weight estimation.  I suggest that the algorithms in this article use the following format: (see attached file).

Thank you for your comment. We have added a pseudo-code of GBDT training with fusion features for broiler weight estimation in the article(L428-429). Thank you again for your valuable advice.

ALGORITHM: GBDT training with fusion features for broiler weight estimation

INPUT:     S = {xi, yi}:Training set, xi is fusion features with 2,073 dimension,yi

is body weight

            K:The number of regression decision trees

           ρ:The learning rate

OUTPUT:  FK(x):Trained gradient boosting regression tree

1. Set squared loss function:

2. Initializ ,whereis the average weight, and xi is the ith sample.

3. for each tree k in K do:

4.    Calculate the negative gradient according to the loss function.

5.    Establish a binary regression decision tree with J leaf region.

6.    Calculate the predicted weight of each.

7.    Get the prediction of the kth regression decision tree and update result.

8. end for

11.Very good bibliography. I hope you can consult more bibliography.

Thank you for your comment. We added and quoted the following 5 articles on weight estimation of pigs, cattle, and sheep and 2 articles on image processing:

Dohmen, R.; Catal, C. & Liu, Q. Computer vision-based weight estimation of livestock: a systematic literature review. New Zealand Journal of Agricultural Research 2022, 65, 227-247.

Ruchay, A.N.; Kober, V.; Dorofeev, K.A.; Kolpakov, V.; Dzhulamanov, K.M.; Kalschikov, V.V.; Guo, H. Comparative analysis of machine learning algorithms for predicting live weight of Hereford cows. Computers and Electronics in Agriculture 2022, 195, 106837.

Gjergji, M.; et al. Deep Learning Techniques for Beef Cattle Body Weight Prediction. In Proceedings of the International Joint Conference on Neural Networks (IJCNN), Glasgow, UK, 19-24 July 2020; pp. 1-8.

Kwon, K., Park, A., Lee, H., & Mun, D. Deep learning-based weight estimation using a fast-reconstructed mesh model from the point cloud of a pig. Computers and Electronics in Agriculture 2023, 210, 107903.

Mahmud, M. S., Zahid, A., Das, A. K., Muzammil, M., & Khan, M. U. A systematic literature review on deep learning applications for precision cattle farming. Computers and Electronics in Agriculture 2021, 187, 106313.

García-Mateos, G., Hernández-Hernández, J. L., Escarabajal-Henarejos, D., Jaén-Terrones, S., & Molina-Martínez, J. M. Study and comparison of color models for automatic image analysis in irrigation management applications. Agricultural water management 2015, 151, 158-166. 

Hernández-Hernández, J. L., García-Mateos, G., González-Esquiva, J. M., Escarabajal-Henarejos, D., Ruiz-Canales, A., & Molina-Martínez, J. M. Optimal color space selection method for plant/soil segmentation in agriculture. Computers and Electronics in Agriculture 2016, 122, 124-132. 

Round 2

Reviewer 1 Report

Comments and Suggestions for Authors

I formulated the following comments to the previous version of the article:

1. The title of the article contains the term “decision tree”, but as such this term appears mainly in section 2.6. The question arises: how do authors classify images? Why is “decision tree” not implemented as such?

2. Here the authors will of course tell me that “decision tree” is GBDT, and enough has been written about this GBDT in the article. BUT not everything is so simple. The fact is that the authors do not use GBDT, but someone else’s (framework). There is nothing wrong with this, it’s just that in this case there is no scientific novelty. So what is the scientific novelty of your research, dear authors?

3. You cannot talk about the use of machine learning or artificial intelligence in isolation from data. And in the article, the authors specifically mention “our data.” The problem is that there is no link through which I could familiarize myself with this dataset. In addition, the authors can prove that their dataset is representative for training, for example, the mentioned R-CNN.

4. By the way, when mentioning R-CNN, I will not forget to say that this type of neural networks suffers from myopia. These neural networks are good at recognizing small details in images (which is why they are often used to detect contours), but they are bad at perceiving the picture as a whole. Transformers do the latter well. So complex transformer+CNN neural networks are now popular. Why don't authors use this?

5. The authors solve the classification problem and evaluate the results in metrics (5)-(8). This is a good, proven metric, but for a regression problem. Where are the errors of the first and second types, dear authors?The authors responded to all my comments. I was not convinced by their answers. Unfortunately, this is one of the frequent articles now where the authors try to pass off the use of a somehow suitable neural network model as scientific novelty, required for research articles. This is the same as trying to pass off the text of a bachelor's thesis as a PhD dissertation. I cannot recommend this article for publication until the authors show significant scientific novelty (analytical models, mathematical justification for changes in the logic of AI operation). Now this is a progress report. In this case, authors can simply change the work type from Article to Report.

Author Response

Q1: The authors responded to all my comments. I was not convinced by their answers. Unfortunately, this is one of the frequent articles now where the authors try to pass off the use of a somehow suitable neural network model as scientific novelty, required for research articles. This is the same as trying to pass off the text of a bachelor's thesis as a PhD dissertation. I cannot recommend this article for publication until the authors show significant scientific novelty (analytical models, mathematical justification for changes in the logic of AI operation). Now this is a progress report. In this case, authors can simply change the work type from Article to Report.

A1: Thank you for your comments. It is economically important to obtain the weight information of broilers in chicken farms, as weight is a crucial indicator of the broiler growth. In intelligent poultry farms, broiler weight is closely related to broiler yield, which most directly reflects their growth. Meanwhile, it is essential to monitor, record, and predict broilers’ body weight for rational intervention in their feed intake and health management, so that the optimal market timing can be determined. In addition, broiler weight information can be used to fit the growth curve, reveal the growth pattern of broilers, and estimate the weight uniformity for efficient flock management.

        Traditional manual weight estimation methods are not only time-consuming and labor-intensive but also have the potential to induce stress reactions in broiler chickens(L59-L73). Furthermore, existing research has predominantly focused on weight measurements for young broiler chickens, with little attention given to broilers aged 42 days or older. Existing research findings suggest that as broiler chickens grow older, the weight estimation error increases. This is due to the less uniform growth of the chicken population and the increased crowding within broiler houses(L124-L131).

        As the largest poultry farming company in Asia, Wens Foodstuff Group is actively driving digital transformation to achieve more efficient management of poultry population. In order to meet this need, we collaborated with Wens Foodstuff Group technical team (co-author Shikai Sun and Huanlong Xie) to jointly develop a non-invasive, automated method for estimating chicken weight, known as MFF-GBDT(see Figure 1 in "Reply  to Reviewer 1.pdf").

        Our approach begins with the design of a set of devices (L173-L178) that utilize depth cameras to capture depth images of chicken flocks. Next, we segment the outlines of the chickens from these depth images (L227-L263). Subsequently, we employ a unique method to fuse artificial 2D features and depth 3D image features from chicken images (L264-L400), enhancing the accuracy of weight estimation. This feature fusion process is not a simple stacking of modules; instead, it is a meticulously designed method that seamlessly integrates features from various sources.

        Ultimately, we feed these fused features into a GDBT (See pseudo code in L428- L429) model for regression-based weight prediction (L401-L429). This innovative approach enables us to extract weight information from depth images, achieving an end-to-end weight estimation.

        Our method does not rely on straightforward component stacking, does not involve image classification, and is not dependent on a single model such as Mask-RCNN or GDBT. Instead, our research begins with the acquisition of depth images and goes through a series of unique feature fusion and regression prediction. In the end, it successfully realizes the estimation of chicken weight, providing an efficient solution for poultry management within Wens Foodstuff Group.

        In summary, our innovation is evident in several key areas:

  1. We have introduced the MFF-GBDT framework, which is capable of non-invasively, efficiently, and accurately estimating the weight of broiler chickens. Moreover, it performs exceptionally well with older broiler chickens. The segmentation of individual chicken images takes an average of only 0.155 seconds. The Mean Absolute Error (MAE) stands at 0.093034 kg, and the R-squared (R2) value is 0.706968, surpassing the existing methods. This framework is currently being implemented in practical application within the industry.
  2. For high-value broiler chickens, we have leveraged the prior knowledge of poultry experts from Wens Foodstuff Group to design and extract uniquely effective artificial 2D features from the depth images of chickens, such as Projected Area (as detailed in L339-L340). We have also introduced a distinctive feature fusion method, seamlessly combining artificial 2D features with depth 3D features. This integration effectively reduces weight estimation errors.
  3. We have made our code open-source and provided a depth image dataset that we created ourselves (https://github.com/GoldfishFive/MFF-GBDT). This dataset is suitable for estimating the weight of older chickens. It was painstakingly collected with a substantial investment from Wens Foodstuff Group. Our intention is to contribute to the field of chicken weight estimation research and development by sharing these valuable resources. Additionally, our image capture devices can easily be transformed into portable handheld devices for practical applications. Furthermore, the improvements we made can be universally applied to weight estimation for various other poultry species.

Q2: The title of the article contains the term “decision tree”, but as such this term appears mainly in section 2.6. The question arises: how do authors classify images? Why is “decision tree” not implemented as such?

A2: Thank you for your comments. Gradient boosting algorithm is a Machine learning technology for regression, classification and ranking tasks, and the learner based on Gradient Boosting algorithm is called GBM(Gradient Boosting Machine). In theory, GBM can choose a variety of different learning algorithms as base learners. When the basis learner we use is a Decision Tree, then the Gradient Boosting algorithm is the gradient Boosting decision Tree GBDT(Gradient Boosting Decision Tree), also known as MART (Multiple Additive Regression Tree). It is an iterative decision tree algorithm, which consists of multiple decision trees, and the conclusions of all trees add up to get the predicted weight. I sincerely hope that the above explanation can be helpful.

Q3: Here the authors will of course tell me that “decision tree” is GBDT, and enough has been written about this GBDT in the article. BUT not everything is so simple. The fact is that the authors do not use GBDT, but someone else’s (framework). There is nothing wrong with this, it’s just that in this case there is no scientific novelty. So what is the scientific novelty of your research, dear authors?

A3: Thank you for your comments. In summary, our innovation is evident in several key areas:

  1. We have introduced the MFF-GBDT framework, which is capable of non-invasively, efficiently, and accurately estimating the weight of broiler chickens.
  2. For high-value broiler chickens, we have leveraged the prior knowledge of poultry experts from Wens Foodstuff Group to design and extract uniquely effective artificial 2D features from the depth images of chickens, such as Projected Area (as detailed in L339-L340). We have also introduced a distinctive feature fusion method, seamlessly combining artificial 2D features with depth 3D features. This integration effectively reduces weight estimation errors.
  3. We have made our code open-source and provided a depth image dataset that we created ourselves (https://github.com/GoldfishFive/MFF-GBDT). This dataset is suitable for estimating the weight of older chickens.

Q4: You cannot talk about the use of machine learning or artificial intelligence in isolation from data. And in the article, the authors specifically mention “our data.” The problem is that there is no link through which I could familiarize myself with this dataset. In addition, the authors can prove that their dataset is representative for training, for example, the mentioned R-CNN.

A4: Thank you for your comments. We have made our code open-source and provided a depth image dataset that we created ourselves (https://github.com/GoldfishFive/MFF-GBDT). This dataset is suitable for estimating the weight of older chickens. It was painstakingly collected with a substantial investment from Wens Foodstuff Group. Our intention is to contribute to the field of chicken weight estimation research and development by sharing these valuable resources. Additionally, our image capture devices can easily be transformed into portable handheld devices for practical applications. Furthermore, the improvements we made can be universally applied to weight estimation for various other poultry species.

Q5: By the way, when mentioning R-CNN, I will not forget to say that this type of neural networks suffers from myopia. These neural networks are good at recognizing small details in images (which is why they are often used to detect contours), but they are bad at perceiving the picture as a whole. Transformers do the latter well. So complex transformer+CNN neural networks are now popular. Why don't authors use this?

A5: Thank you for your comments. Our paper focuses on feature extraction. Segmentation is just a process of image preprocessing, and its purpose is to segment the target for subsequent feature extraction. To segment broiler instances, we use an instance segmentation model, Mask R-CNN, instead of the target detection model, R-CNN. Since the broiler segmentation task is not difficult, using the simple convolutional model Mask R-CNN is more efficient than the complex transformer+CNN model. It is worth mentioning that we extracted the learned features using a simple convolutional model called ResNet50, which is widely used in the community. Many thanks again.

Q6:The authors solve the classification problem and evaluate the results in metrics (5)-(8). This is a good, proven metric, but for a regression problem. Where are the errors of the first and second types, dear authors?

A6: Thank you for your comments. This is not a classification problem, and it is a regression problem. Our task is to predict the weight of broilers rather than the weight classification. The metrics (5)-(8) are important measures of regression error (between true weight and predicted weight of broiler). The errors of the first and second types do not relate to our problem.

Reviewer 2 Report

Comments and Suggestions for Authors

The manuscript was well revised.

Author Response

Thank you for your comments.

Reviewer 3 Report

Comments and Suggestions for Authors

I thank the authors for addressing all the comments on the article.

Author Response

Thank you for your comments.